# How much off-the-shelf knowledge is transferable from natural images to pathology images?

**Xingyu Li** [1]*, **Konstantinos N. Plataniotis** [2]

**1** Electrical and Computer Engineering, University of Alberta, Edmonton, AB, Canada, **2** The Edward S. Rogers Department of Electrical and Computer Engineering, University of Toronto, Toronto, ON, Canada

* xingyu.li@ualberta.ca

**Data Availability Statement:** All datasets used in this manuscript are publicly accessible. IIT Breast cancer image set: (http://www.cs.technion.ac.il). ICIAR2018 grand challenge on breast cancer

## Abstract

Deep learning has achieved a great success in natural image classification. To overcome data-scarcity in computational pathology, recent studies exploit transfer learning to reuse knowledge gained from natural images in pathology image analysis, aiming to build effective pathology image diagnosis models. Since transferability of knowledge heavily depends on the similarity of the original and target tasks, significant differences in image content and statistics between pathology images and natural images raise the questions: how much knowledge is transferable? Is the transferred information equally contributed by pre-trained layers? If not, is there a sweet spot in transfer learning that balances transferred model's complexity and performance? To answer these questions, this paper proposes a framework to quantify knowledge gain by a particular layer, conducts an empirical investigation in pathology image centered transfer learning, and reports some interesting observations. Particularly, compared to the performance baseline obtained by a random-weight model, though transferability of off-the-shelf representations from deep layers heavily depend on specific pathology image sets, the general representation generated by early layers does convey transferred knowledge in various image classification applications. The trade-off between transferable performance and transferred model's complexity observed in this study encourages further investigation of specific metric and tools to quantify effectiveness of transfer learning in future.

## Introduction

Pathology is a medical sub-specialty that studies and practices the diagnosis of disease through examining biopsy samples under microscopes by pathologists. It serves as the golden truth of cancer diagnosis. To address subjectivity in pathology examination [1, 2], computational pathology exploits image analysis and machine learning for histological information understanding in tissue images. Owing to its time-efficiency, consistency, and objectivity, computational pathology merges as a promising approach to cancer diagnosis and prognosis. Inspired by domain knowledge of cancer diagnosis, many algorithms based on hand-crafted feature

histology images: (https://iciar2018-challenge.grand-challenge.org/).

**Funding:** The author(s) received no specific funding for this work.

**Competing interests:** The authors have declared that no competing interests exist.

engineering were proposed to classify pathology images using nuclei's morphology and spatial-distribution features and image texture features [3–10]. Though pathology image diagnosis has achieved impressive progress using hand-crafted feature engineering, effective numerical representation of heterogeneous histological information in pathology images is still the bottleneck. To address this issue, data-driven methods, especially the end-to-end training of convolutional neural network (CNN), are adopted more often in recent pathology image classification studies [11–18]. Though data sets containing hundreds of pathology images are considered "quite" large, they are still far smaller compared to the number of parameters in a medium-size neural network. Consequently, deep diagnostic models training with these data sets are prone to over-fitting and less generalizable in pathology practice.

To address the shortage of large database in deep pathology learning, collecting large pathology image set is highly desirable. However, due to difficulty and time-consuming nature of pathology annotation, large pathology databases with labels are expensive to collect. With recent advance in whole-slide imaging, we believe that very large pathology image sets would accelerate the development of deep learning in computational pathology. At the same time, alternative candidate to address the shortage of large database in deep learning is transfer learning. In transfer learning, a "data-hungary" net is first trained on a very large database, e.g. ImageNet, and the pre-trained model is then applied to relevant but different tasks. Many studies have demonstrated its effectiveness in data-scarce applications related to natural image classification and object recognition [19–22], and natural language processing (NLP) [23]. However, due to the lack of very large annotated pathology image database, there is no reliable pre-trained deep model available in computational pathology. Hence, different from prior studies where data in the original and target tasks share similar properties (e.g. training and test sets are composed of natural images), transfer learning in computational pathology usually adopts pre-trained CNNs on natural images instead [24–29].

It should be noted that though there are different strategies, transfer learning is essentially the use of knowledge gained in one task to solve a new but related problem. Hence, transferability of knowledge heavily depends on the similarity between original and target tasks, and features transfer more poorly when the datasets are less similar [21]. Consequently, on one hand, when using off-the-shelf features in transfer learning, one needs to identify the layers generating general features so that layers computing task-specific features are either discarded or fine-tuned; On the other hand, in the transfer learning strategy of fine-tuning a pretrained model, one needs to specify the values of hyperparameters in finetuning, such as the learning rate and the number of iterations for model refinement (i.e. similar target and source tasks usually requires less refinement). As researchers focusing on computational pathology, we are fully aware the significant differences in image contents and statistics between pathology images and natural images (which is demonstrated in Fig 1), and want to investigate effectiveness of transfer learning by answering following questions:

- Is transfer learning still effective from natural image classification to computational pathology?

- Which layer in a deep net contributes more to pathology image diagnosis?

- Is there a sweet spot to balance transferred model's complexity and performance?

Though answers to these questions form the basis of current pathology-image centered transfer learning, seldom literature tackles them explicitly and, to the best of our knowledge, there are only two studies related to our questions. The study in [26] concludes that fine-tuning a pre-trained net outperforms training a CNN from scratch in medical image analysis.

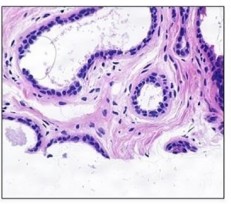 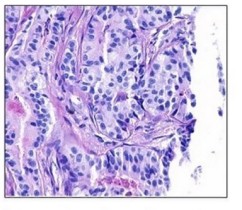 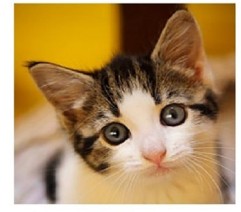 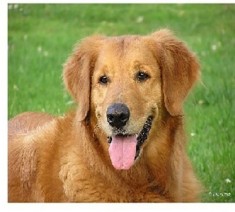

(a) Normal breast tissue     (b) Invasive ductal carcinoma          (c) Cat                    (d) Dog

**Fig 1. Examples of hemotoxylin and eosin (H&E) stained pathology images and natural images.** Image (a) corresponds to a normal tissue while image (b) contains abnormal breast cancer tissue. Compared to the natural images (c)-(d), pathology images containing normal tissues and cancerous tumor appears more similar.

However the experimentation does not include pathology image sets. Recently, different strategies to combine off-the-shelf features are investigated in pathology image centered transfer learning [29]. Since this study focuses on comparison of different pre-trained models (i.e. VGG16, ResNet, and DenseNet et al.), it is non-trivial to infer the descriptive power of off-the-shelf representations by layers directly from its results. In addition, neither of them discuss the trade-off between transferred model's complexity and performance.

## Our contributions

To answer above questions, we define a framework to measure information gain of a particular layer in a pre-trained CNN. Using performance of a random-weight layer as the comparison baseline, the knowledge gain of that particular layer is quantified by the gap between their classification accuracy. We conduct experimentation using two public-accessible breast cancer pathology image sets in this study. Based on the experimental results, though middle-layer representations lead to the highest diagnosis rates, we observe that (i) transferred general knowledge mainly resides in early layers, (ii) the depth layers in a CNN may bring marginal performance improvement in transfer learning, but the complexity of the transferred model (i.e numbers of parameters) increased greatly. This trade-off between transferred model's complexity and transferable performance encourages further investigation of specific metric and tools to quantify effectiveness of transfer learning in future. Note, though fine-tuning a pre-trained model may achieve better performance over the strategy of extracting off-the-shelf representation, the focus of this study is the amount of knowledge that can be reusable in the pretrained net. In addition, fine-tuning a model requires larger data set. Considering data scarcity in current computational pathology research, this study focuses on investigation of off-the-shelf feature extraction methods only.

The rest of this paper is organized as follows. The proposed method to measure knowledge gain of a particular layer in transfer learning is presented in the Methodology Section. Experimental results and discussions are presented in the Experimentation Section, followed by conclusions.

## Methodology: Framework to measure reusable knowledge in transfer learning

In deep learning, the incremental learning nature ensures the transition of representations in layers from generality to specificity. Hence, to reuse a model to a new task, one needs to know how much knowledge is reusable and thus to identify the layers that generate general features, or to specify hyper-parameters in model's fine-tuning. To investigate the amount of reusable

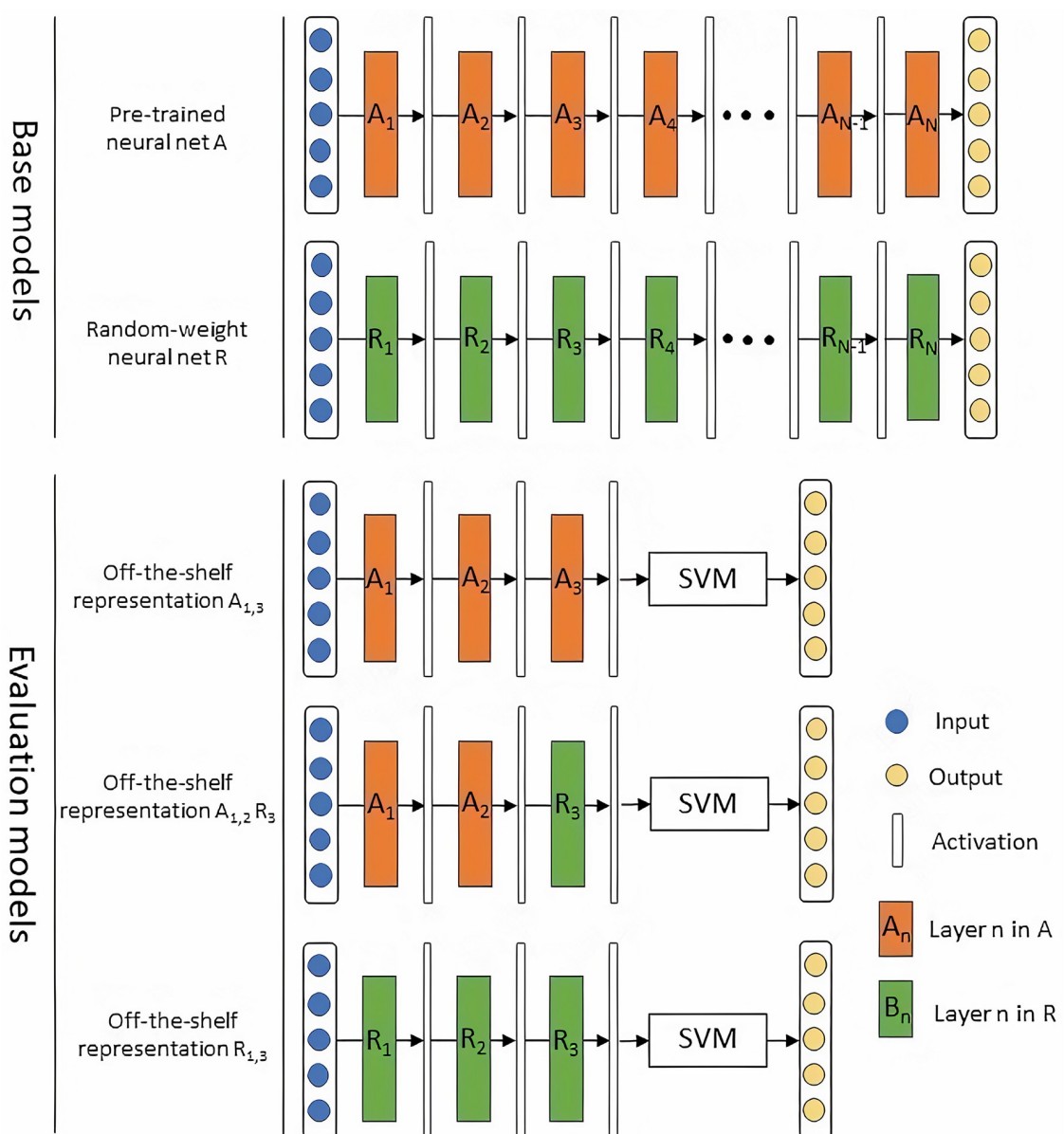

**Fig 2. Overview of the evaluation method for knowledge gain in off-the-shelf features.** In the two base models, model *A* is pre-trained on natural images and net *R* is composed of random-weight layers. Three evaluation models are defined to measure knowledge gains in transfer learning. In this figure, we use layer *n* = 3 as the example layer chosen. The performance difference between models $A_{1,3}$ and $A_{1,2} R_3$ are contributed by knowledge transferred from the third layer of the pre-trained model, $A_3$. And the overall information gained by the first 3 layers of the pre-trained model is quantified by the performance difference between $A_{1,3}$ and $R_{1,3}$.

knowledge in transfer learning, we define a framework to measure the knowledge gain in each layer of a pre-trained net.

Specifically, as presented in Fig 2, we first define two base models. Assume that a CNN *A* has been trained using a database in the original task $T_A$. Its off-the-shelf features are extracted from different layers and passed to a support vector machine (SVM) for a new task $T_B$. Following the identical architecture of *A*, we define a neural network *R* with all convolutional and fully connected layers having random weights. In this figure, layer *n* in the pre-trained model

is denoted by $A_n$; Similarly, random-weight layer $n$ in the model $R$ is represented by $R_n$. The labeled color rectangles (e.g. $A_1$ and $R_1$) represent the weight vectors for that layer, with color differentiating the pretrained and random weights. The vertical transparent bars between weight vectors represent activations at each layer. Then to evaluate the amount of knowledge transferred by the off-the-shelf representation in layer $A_n$, we build three models based on the two base nets as follows:

1. $R_{1,n}$ + SVM: numerical features generated by the first $n$ layers in the random-weight model $R$ are passed to a SVM classifier. Its performance constitutes the comparison baseline in this study.

2. $A_{1,n}$ + SVM: the first $n$ layers of the pre-trained model $A$ are used to compute the off-the-shelf representation. The obtained features are then passed to a SVM machine. The performance gain to the comparison baseline is the overall knowledge gain transferred by the first $n$ layer in model $A$.

3. $A_{1,n-1} R_n$ + SVM: the first $n-1$ layers in model $A$ concatenating with the $n^{th}$ layer in model $R$ are used to generate features for the target task $T_B$. The performance difference between $A_{1,n}$ and $A_{1,n-1} R_n$ are the information gain obtained by the $n^{th}$ layer of model $A$.

In the following sections of this paper, we name the three models $R_{1,n}$, $A_{1,n}$, and $A_{1,n-1} R_n$ for short.

In summary, given a pre-trained model $A$ and a target task $T_B$, we measure the quantity of transferred knowledge in $A$ by comparing its performance to net $R$'s performance in task $T_B$. We select a net composed of random-weight layers as a comparison baseline for the following reason. It is reported that the combination of random-weight convolutional layer, relu layer, pooling layer, and normalization layer might achieve similar performance as learned features [30]. Since a random-weight layer knows nothing about both the original and target tasks, its activations deliver knowledge gained without any effort/train. Through comparing the performance of $R_{1,n}$ and $A_{1,n}$, we can tell how much knowledge obtained by the first $n$ layer in model $A$ is transferable to the target task $T_B$. Similarly, the performance difference of $A_{1,n-1} R_n$ and $A_{1,n}$ is attributed to the information brought by layer $A_n$. We repeat the comparison for all $n \in [1, N]$.

## Experimentation

**Data sets.** This experiment quantifies the transferability of off-the-shelf representation by the performance of pathology image classification. The two public pathology images exploited in the study are described as follow. The breast cancer benchmark biopsy dataset collected from clinical samples was published by the Israel Institute of Technology (IIT data set in short) [31]. The image set consists of 361 samples, of which 119 were classified by a pathologist as normal tissue, 102 as carcinoma in situ, and 140 as invasive carcinoma. The samples were generated from patients' breast tissue biopsy slides, stained with H&E. They were photographed using a Nikon Coolpix 995 attached to a Nikon Eclipse E600 at magnification of 40× to produce images with resolution of about 5 $\mu m$ per pixel. No calibration was made, and the camera was set to automatic exposure. The images were cropped to a region of interest of $760 \times 570$ pixels and compressed using the lossy JPEG compression. The resulting images were again inspected by a pathologist to ensure that their quality was sufficient for diagnosis. Fig 3 presents examples of pathology images in this breast cancer benchmark.

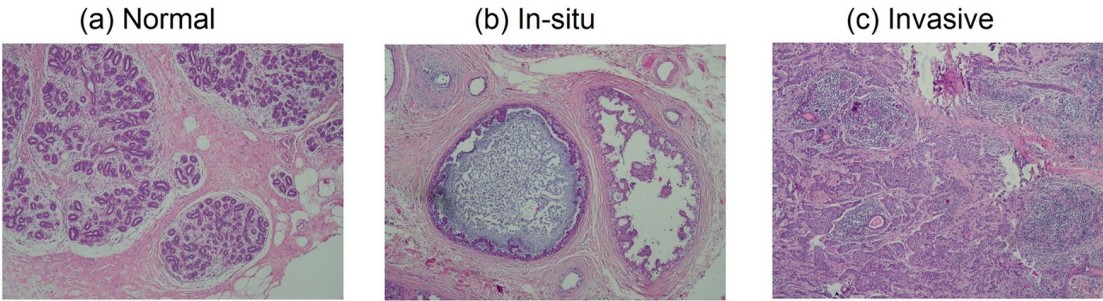

**Fig 3. Examples of pathology images in the IIT benchmark [31].** Images from left to right correspond to normal breast tissue, in-situ breast carcinoma, and invasive breast cancer, respectively.

The second dataset is from the ICIAR2018 Grand Challenges on breast cancer histology images (BATCH) [32]. It is composed of 400 high-resolution (2048 × 1536 pixels) annotated H&E stained images with four balanced classes: normal, benign, in situ carconima and invasive carcinoma. All images are digitized with the same acquisition conditions, with magnification of 200× and pixel size of 0.42 $\mu m$ × 0.42 $\mu m$. Examples of ICIAR2018 image set are shown in Fig 4.

**Deep net architecture.** Considering the experimental datasets have relatively small number of pathology images, we selects the AlexNet (which has fewer layers and parameters compared to other deep models) [33] pre-trained on the ImageNet database as the model $A$ in this experimentation. AlexNet is composed of 25 layers, including 5 convolutional layers and 3 fully-connected layers. In this study, the off-of-shelf features are extracted after the 8 learned layers as illustrated in Table 1. The random-weight neural network $R$ shares the identical architecture as AlexNet but with filter weights randomly generated following the standard normal distribution $N(0, 0.01)$, i.e. Gaussian distribution with zero mean and standard deviation of 0.01.

**Evaluation protocol.** The image set is divided into training set and test set, with a ratio of 7:3. Images in the training set are augmented by rotation with an angle randomly drawn from [0, 360) degrees, vertical reflection, and horizontal flip. The augmented training images are fed to the three evaluation models $A_{1,n}$, $A_{1,n-1} R_n$, and $R_{1,n}$, generating three different feature sets for each $n \in [1, 8]$. Then for each off-the-shelf feature set, a linear SVM is trained and optimized for pathology image diagnosis. In the testing phase, test images are processed by the evaluated models and classified by corresponding linear SVMs. Finally, agreement of classification results and annotated image labels is recorded for comparison. This study uses classification accuracy $ACC \in [0, 1]$ to measure pathology image diagnosis performance. Since the number of images in each category of both datasets is quite close, the limitation of $ACC$ (i.e.

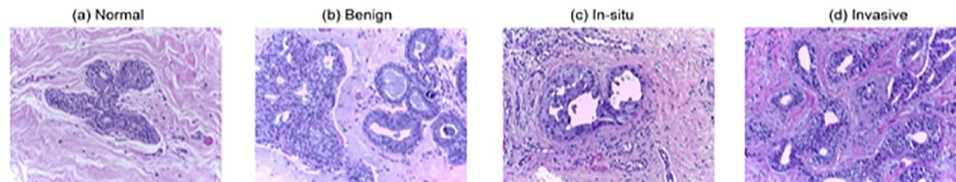

**Fig 4. Examples of ICIAR2018 pathology images [32].** Images from left to right correspond to normal breast tissue, benign tumor, in-situ breast carcinoma, and invasive breast cancer, respectively.

**Table 1. Off-the-shelf feature extraction from AlexNet.** AlexNet is composed of 25 layers, including 5 convolutional layers and 3 fully-connected layers. In this study, the off-of-shelf features are extracted after the 8 learned layers.

| Layer | Layer type | Off-the-shelf features |
|:-----:|:----------:|:----------------------:|
| 0 | input | |
| 1 | convolution | representation layer 1 |
| 2 | Relu | |
| 3 | normalization | |
| 4 | max-pooling | |
| 5 | convolution | representation layer 2 |
| 6 | Relu | |
| 7 | normalization | |
| 8 | max-pooling | |
| 9 | convolution | representation layer 3 |
| 10 | Relu | |
| 11 | convolution | representation layer 4 |
| 12 | Relu | |
| 13 | convolution | rrepresentation layer 5 |
| 14 | Relu | |
| 15 | max-pooling | |
| 16 | fully-connected | representation layer 6 |
| 17 | Relu | |
| 18 | dropout | |
| 19 | fully-connected | representation layer 7 |
| 20 | Relu | |
| 21 | dropout | |
| 22 | fully-connected | representation layer 8 |
| 23 | Softmax | |
| 24 | Output | |

biased by disease prevalence) is mitigated. To obtain a reliable conclusion, we repeat the experiments 50 times for each $n \in [1, 8]$ and obtain the final data by averaging all $ACCs$.

## Results and discussion

The experimental results for the pathology image datasets are shown in Fig 5, where ach marker is the figure represents the average accuracy over the validation set for 50 times. The blue line connects models used off-the-shelf representation $A_{1,n}$ extracted from the $n^{th}$ layer. The Orange line connects models $A_{1,n} R_n$, which applies a random-weights filter layer to the $A_{1,n-1}$ representation, and the gray solid line corresponds to the performance associated with random-weight layer models $R_{1,n}$. Note that for the IIT image set, classification accuracy achieved by the state-of-the-art hand-crafted method [7] is marked by the gray dash line in the left figure for reference. Since no hand-crafted method specifically designed for the BATCH set, gray dash line is not shown in the right figure.

First, for the binary classification of the IIT image set reported on the left of the Fig 5, transfer learning outperforms the hand-crafted method. Then let's focus on $A_{1,n}$ and $A_{1,N-1} R_n$, which are denoted by the blue and orange lines, respectively. The difference between these two models is whether weights in the $n^{th}$ layer are pre-trained. The performance gap is mainly attributed to knowledge transferred from natural image classification to pathology image diagnosis. In this experiment on the IIT image set, most transferable information is delivered by

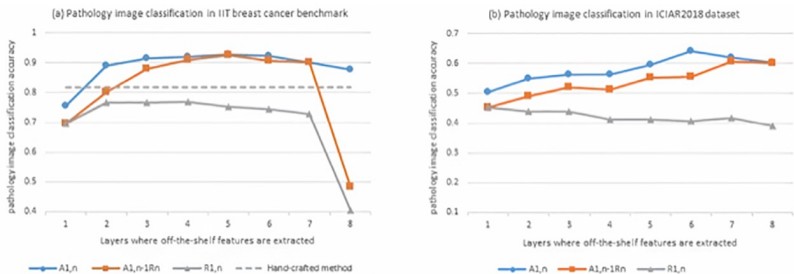

**Fig 5. Transfer learning performance over the pathology image sets.** Each marker is the figure represents the average accuracy over the validation set for 50 times. The blue line connects models used off-the-shelf representation $A_{1,n}$ extracted from the $n^{th}$ layer. The Orange line connects models $A_{1,n} R_n$, which applies a random-weights filter layer to the $A_{1,n-1}$ representation, and the gray solid line corresponds to the performance associated with random-weight layer models $R_{1,n}$. For reference, classification accuracy achieved by the state-of-the-art hand-crafted method [7] is marked by the gray dash line. As we propose in this study, the knowledge gain of the $n^{th}$ layer can be quantified by the performance difference between $A_{1,n}$ and $A_{1,n-1} R_n$ and the classification difference between $A_{1,n}$ and $R_{1,n}$ represents how much knowledge is transferable in first $n$ layers in the pre-trained CNN.

the first and second layers and increase of layer index comes with marginal performance improvement after the third layer. Performance difference between the blue line $A_{1,n}$ and the gray solid line $R_{1,n}$ reveals total amount of transferable information accumulated by the first $n$ layers in the pre-trained AlexNet. Performance gap grows slightly wider from layer $n = 3$ to $n = 6$. This observation again verifies that the transferred middle layers in the pre-trained model do not introduce more knowledge compared to the random-weights layers $R_{1,n}$ for $3 \leq n \leq 6$. Above observations suggests that applying the first two layers in the pretrained AlexNet to IIT image classification is the sweet point to balance the classification performance and model's complexity.

The BATCH image set poses a problem of 4-category pathology image classification. In the left figure of Fig 5, we observe a steady increment of diagnosis accuracy from the first layer to the sixth layer. Transferring the fully-connected layers in the representation layer 7 and 8 degrades the diangosis performance. Compared to the experiment on the IIT image set, the sweet spot for model transfer (i.e. transferring representation layer 1 to 6) is more obvious. Since effectiveness of transfer learning depends on a specific image set, it encourages the further investigation of specific metric and tools to quantify the feasibility of transfer learning in future.

## Conclusions

In this work, we proposed a framework to quantify the amount of information gained by each pre-trained layer, and experimentally investigated and reported transfer efficiency of deep net's off-the-shelf representation over different pathology image sets. The experiments suggested that the off-the-shelf features learned from natural images can be reused in compuational pathology, but the amount of information that could be transferable heavily depended on complexity of pathology images. The observation in this study had practical reference to pathology image centered transfer learning.

## Author Contributions

**Conceptualization:** Xingyu Li.

**Formal analysis:** Xingyu Li.

**Methodology:** Xingyu Li, Konstantinos N. Plataniotis.

**Software:** Xingyu Li.

**Writing – original draft:** Xingyu Li, Konstantinos N. Plataniotis.

**Writing – review & editing:** Xingyu Li, Konstantinos N. Plataniotis.

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
