## [Decision Letter · Decision Letter 0]

18 Aug 2020

PONE-D-20-16898

How Much Off-The-Shelf Knowledge Is Transferable From Natural Images To Pathology Images?

PLOS ONE

Dear Dr. Li,

Thank you for submitting your manuscript to PLOS ONE. After careful consideration, we feel that it has merit but does not fully meet PLOS ONE’s publication criteria as it currently stands. Therefore, we invite you to submit a revised version of the manuscript that addresses the points raised during the review process.

We look forward to receiving your revised manuscript.

Kind regards,

Tao Song

Academic Editor

PLOS ONE

Journal Requirements:

2. Please revise the typographical error on page 2, line 22: the text should read "data-hungry" not "hungary".

Reviewers' comments:

Reviewer's Responses to Questions

**Comments to the Author**

1. Is the manuscript technically sound, and do the data support the conclusions?

Reviewer #1: Yes

Reviewer #2: Partly

2. Has the statistical analysis been performed appropriately and rigorously? 

Reviewer #1: Yes

Reviewer #2: Yes

3. Have the authors made all data underlying the findings in their manuscript fully available?

Reviewer #1: Yes

Reviewer #2: Yes

4. Is the manuscript presented in an intelligible fashion and written in standard English?

Reviewer #1: Yes

Reviewer #2: Yes

5. Review Comments to the Author

Reviewer #1: The questions raised in the introduction are properly answered in the conclusion. The results are efficient and promising.

However, I don't think the images (c)-(d) in Figure 1 makes sense. Image (a) corresponds to a normal tissue and image (b) contains abnormal breast cancer tissue, while (c) and (d) contain two separate species. In my opinion, natural images of two separate species should be used for comparison, rather than images of the same specie.

Reviewer #2: In the paper, authors focus on transferability of knowledge in deep learning. They define a framework to measure information gain and the experiment results are good. However, there are some details should be noticed.

Since you mentioned there are two studies before, did you compare the results with them?

You should improve the quality of the figures in the paper. Some of them are dim.

Take care of the use of punctuation from line 75 to 78.

6. PLOS authors have the option to publish the peer review history of their article (what does this mean?). If published, this will include your full peer review and any attached files.

Reviewer #1: No

Reviewer #2: No

---

## [Author Response · Author response to Decision Letter 0]

25 Aug 2020

We would like to thank the AE and reviewers for the time taken to both read the document and make criticism on it. Their inputs are well received and taken into consideration in preparing our revision, which has resulted in a manuscript that is clearer and much improved. In addition, we would like to thank the Academic Editor for the time taken to organize this review process. 

This document provides a point-by-point feedback to the concerns raised by AE and reviewers and indicates how these comments haven been incorporated in the revised manuscript. The reviewers’ comments are started with the letter Q. Answers to the reviewers’ questions are started with the letter R. Correspondingly, in the revised marked-up manuscript, all alterations made due to the reviewers’ considerations are highlighted by yellow lettering to ease further reviewing. All references and citations included below corresponds to the revised manuscript. 

AE: When submitting your revision, we need you to address these additional requirements.

Q1: Please ensure that your manuscript meets PLOS ONE's style requirements, including those for file naming. The PLOS ONE style templates can be found at

R1: Point well taken. In light of AE’s comment, the title page and main body of the manuscript are reformatted accordingly.

Q2: Please revise the typographical error on page 2, line 22: the text should read "data-hungry" not "hungary".

R2: Point well taken. We would like to thank the AE for pointing out this typo. 

Review 1: The questions raised in the introduction are properly answered in the conclusion. The results are efficient and promising.

Q1: However, I don't think the images (c)-(d) in Figure 1 makes sense. Image (a) corresponds to a normal tissue and image (b) contains abnormal breast cancer tissue, while (c) and (d) contain two separate species. In my opinion, natural images of two separate species should be used for comparison, rather than images of the same species.

R1: We would like to thank the review for this comment. We agree with the reviewer that using images of separate species for demonstration is better. In Fig 1, we use a cat image (c) and a dog image (d) as examples of natural images. In my opinion, these images do contain objects from separate species.

Review 2: In the paper, authors focus on transferability of knowledge in deep learning. They define a framework to measure information gain and the experiment results are good. However, there are some details should be noticed.

Q1: Since you mentioned there are two studies before, did you compare the results with them?

R1: We would like to thank the review for this comment. Though there are two prior works on evaluating transfer learning performance on medical images, neither of them discusses the transferability of off-the-shelf representations by layers. Because the focuses of prior works and our study are completely different, it is non-trivial to setup an experimental benchmark for result comparison. Specifically, as stated in the original manuscript, 

- the study in [26] compares fine-tuning a pre-trained net and training a CNN from scratch in medical image analysis. Their experimentation does not include any pathology image sets. 

- Recent study in [29] investigates different strategies to combine off-the-shelf features in pathology image centered transfer learning. Since this study focuses on comparison of different pre-trained models (i.e. VGG16, ResNet, DenseNet, et al.), it is non-trivial to infer the descriptive power of off-the-shelf representations by layers directly from its results. 

- Our work proposes a quantitative framework to investigate the transferability of off-the-shelf representations by layers within a pre-trained model. 

In sum, the different themes of the three studies make it difficult to directly compare the results under one experimental setting.

Q2: You should improve the quality of the figures in the paper.

R2: Point well taken. In the revision, we have improved the quality of figures (e.g. Fig 2 and Fig 5). Thank you.

Q3: Take care of the use of punctuation from line 75 to 78.

R3: Point well taken. The structures of the manuscript from line 75-78 has been revised and is reproduced below for ease of review: “The rest of this paper is organized as follows. The proposed method to measure knowledge gain of a particular layer in transfer learning is presented in the Methodology Section. Experimental results and discussions are presented in the Experimentation Section, followed by conclusions.”

---

## [Editor Report · Decision Letter 1]

29 Sep 2020

How Much Off-The-Shelf Knowledge Is Transferable From Natural Images To Pathology Images?

PONE-D-20-16898R1

Dear Dr. Li,

We’re pleased to inform you that your manuscript has been judged scientifically suitable for publication and will be formally accepted for publication once it meets all outstanding technical requirements.

Kind regards,

Tao Song

Academic Editor

PLOS ONE
---

## [Editor Report · Acceptance letter]

5 Oct 2020

PONE-D-20-16898R1 

How much off-the-shelf knowledge is transferable from natural images to pathology images? 

Dear Dr. Li:

I'm pleased to inform you that your manuscript has been deemed suitable for publication in PLOS ONE. Congratulations! Your manuscript is now with our production department. 

Kind regards, 

on behalf of

Dr. Tao Song 

Academic Editor

PLOS ONE